



# Aggregation of ice-nucleating macromolecules from *Betula pendula* pollen determines ice nucleation efficiency.

Florian Wieland[1,2], Nadine Bothen[2], Ralph Schwidetzky[3], Teresa M. Seifried[1,4], Paul Bieber[1,4], Ulrich Pöschl[2], Konrad Meister[3,5], Mischa Bonn[3], Janine Fröhlich-Nowoisky[2], Hinrich Grothe[1]

5    [1]Institute of Materials Chemistry, TU Wien, Vienna, 1060, Austria
     [2]Multiphase Chemistry Department, Max Planck Institute for Chemistry, Mainz, 55128, Germany
     [3]Molecular Spectroscopy Department, Max Planck Institute for Polymer Research, Mainz, 55128, Germany
     [4]Department of Chemistry, University of British Columbia, Vancouver, BC V6T 1Z1, Canada
     [5]Department of Chemistry and Biochemistry, Boise State University, Boise, ID 83725

*Correspondence to*: H. Grothe (hinrich.grothe@tuwien.ac.at)

**Short summary (only for submission/website)**

*Betula pendula* is a widespread birch tree species containing ice nucleation agents that can trigger the freezing of cloud droplets, and thereby alter the evolution of clouds. Our study identifies three distinct ice-nucleating macromolecules (INMs)

15    and aggregates of varying size that can nucleate ice at temperatures of up to -5.4°C. Our findings suggest that these vegetation-derived particles may influence atmospheric processes, weather, and climate stronger than previously thought.

**Key figure**

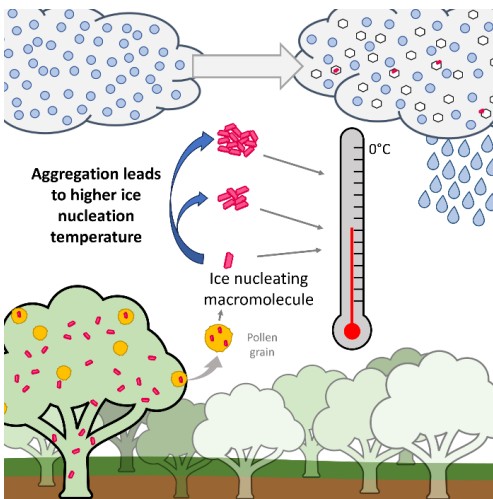

20





**Abstract.**

Various aerosols, including mineral dust, soot, and biological particles, can act as ice nuclei, initiating the freezing of supercooled cloud droplets. Cloud droplet freezing significantly impacts cloud properties and, consequently, weather and climate. Some biological ice nuclei exhibit exceptionally high nucleation temperatures close to 0°C. Ice Nucleating Macromolecules (INMs) found on pollen are typically not considered among the most active ice nuclei. Still, they can be highly abundant, especially for species such as *Betula pendula*, a widespread birch tree species in the boreal forest. Recent studies have shown that certain tree-derived INMs exhibit ice nucleation activity above -10°C, suggesting they could play a more significant role in atmospheric processes than previously understood. Our study reveals three distinct INM classes active at -8.7°C, -15.7°C, and -17.4°C are present in *B. pendula*. Freeze-drying and freeze-thaw cycles noticeably alter their ice nucleation capability, and the results of heat treatment, size, and chemical analysis indicate that INM classes correspond to size-varying aggregates, with larger aggregates nucleating ice at higher temperatures in agreement with previous studies on fungal and bacterial ice nucleators. Our findings suggest that *B. pendula* INMs are potentially important for atmospheric ice nucleation because of their high prevalence and nucleation temperatures.

## 1 Introduction

Heterogeneous ice nucleation is a pivotal process in the Earth's atmosphere as it can influence the properties of clouds. Cloud droplets can be supercooled to −38°C until ice forms homogeneously, but ice-nucleating particles (INPs) can trigger freezing at much higher temperatures. Thereby, INPs influence clouds and their properties, such as lifetime and precipitation. More than half of precipitation initiates via the ice phase of mixed-phase clouds (Lau and Wu, 2003). Therefore, accurate data on INPs, such as composition, concentration, and initial freezing temperatures, is crucial to understanding and accurately modeling their influence on Earth's climate. Various aerosol particles can function as INPs, but the specific properties and prerequisites for nucleating ice are poorly understood. This is particularly true for ice-nucleating primary biological aerosol particles (PBAPs), with certain types active in ice nucleation at relatively high sub-zero temperatures (above −10°C).

Birch pollen is one of the most studied PBAPs and can induce heterogeneous ice nucleation at −15°C (Diehl et al., 2001). In 2012, Pummer et al. found that extractable macromolecules outside the sporopollenin shell of birch pollen are the ice nucleation active substance (Pummer et al., 2012). Field and laboratory studies have shown that these ice-nucleating macromolecules (INMs) are highly abundant in alpine vegetation and originate not only from birch pollen but also from different surfaces of birch trees (Felgitsch et al., 2018; Seifried et al., 2020). During and after precipitation, these INMs were mobilized and aerosolized in high concentrations near mountain forests and above the canopy (Bieber et al., 2020; Seifried et al., 2021). These studies suggest a high abundance of birch INMs but compared to other biological INPs, e.g., of bacterial or fungal origin, which can be active up to −2°C (Corotto et al., 1986; Pouleur et al., 1992; Fröhlich-Nowoisky et al., 2015), birch INMs have relatively low nucleation temperatures (−15°C). Birch INMs have been described as polysaccharides with a molecular weight of 100 and 300 kDa, starting heterogeneous ice nucleation at −15°C (Pummer et al., 2012, 2015; Dreischmeier et al., 2017). Later studies showed that they exhibit typical protein properties (Tong et al., 2015; Felgitsch et al., 2018; Burkart et al., 2021), i.e., denaturation by heat and enzymes. It is still up for discussion whether pollen INMs are saccharides (Pummer et al., 2012; Dreischmeier et al., 2017), proteins (Tong et al., 2015; Felgitsch et al., 2018), or a combination of both. Spectroscopic results supported both findings, showing infrared bands representative of polysaccharides and amides and fluorescence signals associated with proteins (Pummer et al., 2013; Felgitsch et al., 2018; Burkart et al., 2021). Dreischmeier et al. (2017) verified the freezing activity of birch INMs at −15°C and their saccharide nature, but they also published results in the supplement showing ice nucleation activity up to −6°C, which was later attributed to the pollen grains rather than to the INMs (Dreischmeier, 2019). Murray et al. (2022) also reported high freezing temperatures from birch pollen (up to −7.7°C) in large droplet volumes, attributed them to INMs, and hypothesized that freezing temperatures between −6° and −15°C are due to



large aggregates of smaller INMs. Dreischmeier et al. reached a similar conclusion regarding aggregates but in a different context. They suggested that INMs could be aggregates of smaller antifreeze polysaccharides. They based their conclusions on the findings of ice nucleation and antifreeze activity in boreal pollen, and both substances showed similar spectroscopic
signals.

In this study, we focus on (i) the different subpopulations of pollen INP and their initial freezing temperatures, (ii) what influences the ice nucleation activity, and (iii) how aggregation might be responsible for this behavior of birch INMs. Therefore, we used two complementary ice nucleation assays to investigate the different subpopulations and initial freezing temperature differences. We show how simple and common laboratory sample-handling techniques can strongly influence the
ice nucleation activity of pollen INMs. Combined with size measurements, we conclude that aggregation may be a critical property of birch INMs. Finally, we place our results in the context of previous studies and try to contribute to a deeper understanding of birch INMs and a general knowledge of INMs produced by trees.

## 2 Materials & Methods

### 2.1 Sample preparation

Birch pollen from *Betula pendula* (silver birch) was obtained from Pharmallerga (CZ, Czech Republic). According to the distributor, the pollen was harvested by hand in March 2020 in České Budějovice, Czech Republic, hereafter referred to as pollen A, or in March 2019 in Galanta/Trnava/Senec, Slovakia, referred to as pollen B. The individual batches contained pollen from five growing sites (forest, park, land, roadside).

Birch pollen washing water (BPWW), containing birch pollen INMs, was prepared with ultra-high quality (UHQ) water. The
UHQ water was prepared by autoclaving MilliQ water for 20 min at 120°C, followed by filtration through a 0.1 µm sterile PES (polyether sulfone) vacuum filter (three times). Birch pollen and UHQ water were mixed to obtain a 50 mg birch pollen concentration per mL of water. The extraction time was 6 h, during which the mixture was occasionally shaken. The suspension was centrifuged at 350 rpm for 5 min at room temperature, and the supernatant was filtered through a 0.2 µm syringe filter (VWR, Radnor, PA, USA). The BPWW was stored at −20°C if not used immediately.

#### 2.1.1 Sample treatment

For stability testing, BPWW was freeze-dried, freeze-thawed, and heat-treated. Before freeze-drying, BPWW was stored at -80°C for at least 15 minutes. BPWW was dried to a stable mass using an Alpha 2-4 LDplus (Christ, Germany), the duration of which varied according to the volume. Subsequently, water was added to obtain the same mass as before freeze-drying. Freeze-thaw cycles are subsequent measurements with the ice nucleation assay, where the next measurement was started after
all droplets were completely thawed. For heat treatments, aliquots of BPWW (between 750 µL and 1500 µL) were placed in a microtube (1.5 or 2 mL, Eppendorf), sealed with parafilm, and incubated at 40°C, 78°C and 98°C for 1 h or 24 h respectively (Accublock™Mini, Labnet International Inc. USA). After the specified duration, the samples were cooled to RT and analyzed on the same day.

#### 2.1.2 Filtration

For INM size characterization, BPWW was filtered through centrifugal molecular weight cut-off (MWCO) filters with nominal cut-offs at 10, 30, 50, 100, and 300 kDa (Vivaspin©15R/20, Sartorius). The filter material was polyether sulfone (PES) or Hydrosat©(HS) in the case of the 10 kDa filter. The BPWW was filtered at 3000 xg (times gravitational force) for 30 min at 4°C. The filters were rinsed with UHQ water four times before sample filtration to avoid contamination. MWCO filters give nominal cut-off masses; however, these masses correspond to pore sizes and are usually correlated with protein masses. Hence,



applying the results on birch INM mass might be misleading. We used these filters to separate the INMs using the mass as an indicator for size.

### 2.1.3 Purification

Before chemical analysis, the INMs from BPWW were purified by ice-slide ice affinity. Details of the purification method concept have been described elsewhere (Lukas et al., 2021; Marshall et al., 2016; Adar et al., 2018; Schwidetzky et al., 2023).

In brief, the sample is continuously pumped to flow down a metal plate cooled by a cryostat to ~-3.5°C. As the sample slowly flows down, an ice layer forms. About 800 mL BPWW was prepared and diluted with about 2075 mL of UHQ water to obtain a sufficient volume. The plate was not pre-cooled to avoid immediate freezing of the first droplets. After 1 h, the ice sheet was collected, frozen at –20°C and freeze-dried. The dry residue was used to prepare ice-affinity purified BPWW by adding UHQ to a final concentration of 7.2 mg residue mL$^{-1}$. Only one run was performed.

### 2.2 Measurements

#### 2.2.1 Ice nucleation measurements

Two independent droplet-freezing assays were used to investigate the ice nucleation activity in immersion freezing mode. First, the twin-plate ice nucleation assay (TINA) (Kunert et al., 2018) and second, the Vienna optical droplet crystallization analyzer (VODCA) (Pummer et al., 2012; Felgitsch et al., 2018). Procedures for both setups can be found in the respective

publications. The main differences are that the droplets are on a PCR tray in TINA, and in VODCA, the droplets are emulsified in a paraffin-lanolin oil mixture. The cooling rates are very different in 1 K min$^{-1}$ for TINA and 10 K min$^{-1}$ for VODCA. However, previous studies showed that the cooling rate has a negligible effect on immersion freezing temperatures (Wright et al., 2013). TINA uses an IR camera, which detects the latent heat to detect freezing, while VODCA uses a light microscope to detect the difference in optical density (frozen droplets appear darker).

TINA analyzes 98 droplets of 3 µL each and up to 8 samples simultaneously. In contrast, VODCA measures around 100 droplets ranging from 1.8 pL to 33.5 pL. As pL droplets are six orders of magnitude smaller than µL droplets, the probability of homogeneous and heterogeneous freezing events is much higher in µL droplets. Classical nucleation theory suggests smaller droplets, such as those in VODCA, freeze at lower temperatures. The background freezing temperature in VODCA is around -33°C, whereas with TINA, the background freezing of UHQ water has been measured between -20°C and -30°C. In addition,

larger droplets contain more INMs than small droplets. Therefore, the probability of ice nucleation by high-temperature active INMs (present only at low concentration) is increased in large droplets, as the most efficient INM nucleates the whole droplet. Consequently, VODCA and TINA can be considered complementary techniques: VODCA is optimal for high INM concentrations at low temperatures, while TINA is better for low INM concentrations at higher temperatures.

Both methods give a list of freezing points per droplet. In the first step, the fraction of frozen $f_{ice}$ droplets is calculated.

Subsequently, the number of INMs active above a specific temperature $N_m(T)$ can be estimated using Vali's formula (Vali, 1971). Detailed information on the evaluation of data for TINA (Kunert et al., 2018) and VODCA (Seifried et al., 2023) can be found elsewhere. The Gaussian error propagation calculates the uncertainty for each TINA measurement.

#### 2.2.2 Dynamic light scattering

Dynamic light scattering (DLS) was measured using a Nicomp 380 submicron particle sizer (Entegris, USA) with a fixed

scattering angle of 90° and a laser wavelength of $\lambda$ = 632.8 nm. DLS measurements were performed on undiluted ice-affinity purified BPWW samples before and after heat treatment. More details can be found in Schwidetzky et al. (2020, 2021).





### 2.2.3 Circular dichroism spectroscopy

Circular dichroism (CD) spectra were recorded using a J-1500 CD-Spectrometer (Jasco, USA). Samples were measured in a 350 µL quartz cuvette (Hellma Analytics, Germany) with a path length of 1 mm. Spectra were recorded at room temperature
from 190 nm to 250 nm with a data pitch of 0.2 nm, a scan rate of 5 nm min$^{-1}$, and a data integration time of 2 s. Samples were diluted about 1:5 (1 part sample to 5 parts overall) with UHQ water. UHQ Water was measured as background. Spectra were background subtracted and processed using JASCO's Spectra Manager Analysis program.

### 2.2.4 Fluorescence spectroscopy

Fluorescence excitation-emission maps (FEEM) were recorded using an FSP920 spectrometer (Edinburgh Instruments, UK)
with an Xe900 xenon arc lamp (450W) and an S900 single-photon photomultiplier. Samples were measured in a quartz glass cuvette (500 µL, Hellma Quartz (Suprasil©, Germany). Measurement parameters included a dwell time of 0.25 s and monochromator step widths of 5 nm (excitation and emission). First-and second order excitation was prevented using an offset of 10 nm and a 295 nm low-pass filter.

### 2.2.5 Infrared spectroscopy

Fourier-transform-infrared (FTIR) spectroscopic measurements were recorded using a Vertex 80v (Bruker, Germany) with a liquid nitrogen-cooled MCT (mercury cadmium telluride) detector. The spectrometer operates at a pressure of 2 mbar. The internal reflection element (IRE) in the attenuated total reflection (ATR) cell (GladiATR™, Pike Technologies, USA) was a diamond crystal. OPUS 6.5 software (Bruker, Germany) was used for evaluation and instrument control. The crystal surface was flushed with dry nitrogen. For each measurement, 128 scans were accumulated at a resolution of 0.5 cm$^{-1}$. The IRE was
coated with a thin layer of soluble sample components for measurement. Approximately 30 µL of the sample were pipetted onto the IRE and dried with a steady stream of dry nitrogen. Post-processing of the spectra included normalization and removal of the IRE signal between 1900 cm$^{-1}$ and 2350 cm$^{-1}$.

## 3 Results & discussion

### 3.1 Investigating *Betula pendula* INMs using two complementary ice nucleation assays.

Figure 1a shows the ice nucleation activity of BPWW measured with two complementary ice nucleation assays: (i) TINA, which measures 3 µL droplets (Kunert et al., 2018), hereafter referred to as µL-assay, and (ii) VODCA, which measures pL droplets, with an average volume of 8.2 pL (Pummer et al., 2012), hereafter referred to as pL-assay. Figure S1 in the supplement shows background measurements.

The combined results in Figure 1a show the number of nucleation sites ($N_m$) per gram of pollen. Three distinct INMs evolve
in the $N_m$ spectra, which can be recognized via the various rises and plateaus. Starting from the right, a sharp increase between -5.7°C and ~ -7°C highlights the first INM. After a plateau from ~ -7°C to ~ -14°C, the second INM appears between ~ -14°C and ~ -16°C. The third INM follows immediately, triggering freezing between ~ -16°C and ~ -19°C, followed by a plateau at ~ -33°C, after which the water freezes. Hence, we can derive three distinct INM subpopulations with different efficiencies, which, from now on, we term as INM *Classes A, B*, and *C*, with average freezing temperatures of -8.7°C, -15.7°C and -17.4°C,
respectively.

The measurements of the two assays overlap between ~ -15°C and ~ -19°C, but the pL-assay (VODCA) results show only Class *B* and *C* INMs, with the first only being minimally visible. The three INM classes occur in vastly different concentration ranges: *Class A* INMs with a concentration of $N_m < 10^3$ g$^{-1}$ (only detectable with the µL-assay, TINA), C*lass B* INMs with a concentration of $N_m \sim 10^2$ to ~$10^7$ g$^{-1}$, and *Class C* INMs with a concentration of $N_m \sim 10^5$ to ~ $10^{11}$ g$^{-1}$, derived from six



independent measurements (six different BPWW, three prepared with pollen A and three with pollen B) with the µL-assay, see Figure S2 in the supplement.

We used the heterogeneous underlying-based (HUB) stochastic optimization analysis (de Almeida Ribeiro et al., 2023) to better identify and characterize the underlying number of INM subpopulations in pollen. Fig. 1b shows the differential spectrum predicted by HUB from the optimized distribution of nucleation temperatures (e.g., the differential spectrum), shown

in Fig. 1b. The three INM classes in the differential spectrum are centered at the average freezing temperatures of $-8.7$ °C for *Class A* and $-15.7$°C for *Class B* and $-17.4$°C for *Class C*. The differential spectrum shows that *Class B* and *C* INMs combined represent less than 0.01% of the active nucleation sites in pollen.

Over the six independent measurements, concentrations varied, with some containing almost no *Class A* INMs and freezing just above -15°C. INM *Classes B* and *C* also showed substantial variations. Several factors, such as the extraction process and

the natural variability of biological samples, can cause these variations. We evaluated other effects such as different pollen origin (Slovakia and Czech Republic, Pharmallerga) and different storage conditions of the pollen before extraction (room temperature vs. -20°C) and replicated the measurements after one year; none of these factors seemed to influence the ice nucleation activity systematically.

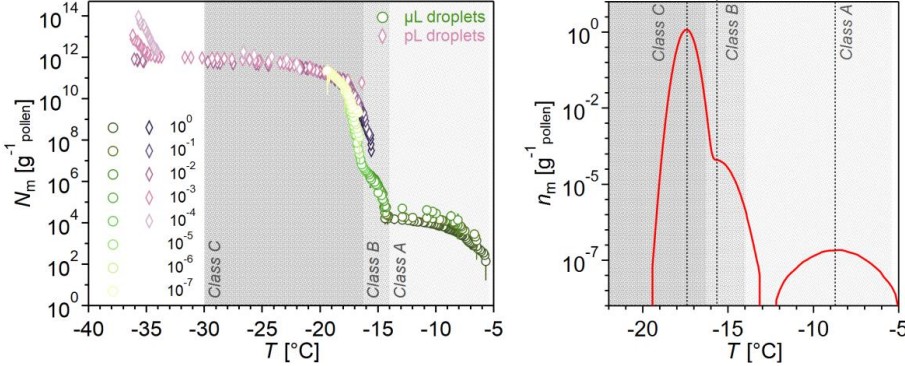

**Figure 1. Freezing spectrum of birch pollen washing water (BPWW) with 50 mg pollen mL$^{-1}$ measured with two complementary ice nucleation assays. (a)** Cumulative number $N_m(T)$ of INMs per gram *Betula pendula* pollen. Green circles show the measurement with the µL-droplet-assay. Pink diamonds represent the measurement with the pL-droplet-assay. The legend on the left shows the dilutions of the sample, e.g., 10$^{-1}$ which corresponds to a 1:10 dilution. The graph shows three distinct average freezing temperatures at -8.7°C, -15.7°C, and -17.4°C marked by the steep increase of the $N_m$ spectra. The first hump (-5.4°C to -14°C) is only measurable with the µL-droplet assay.
The increase at -33°C is the background of the pL-droplet assay. The measurements overlap between -15°C and -20°C. **(b)** Number of INMs $n_m(T)$ analyzed with the heterogeneous underlying-based (HUB) stochastic optimization analysis of de Almeida Riberio et al. (2023) to better identify and characterize the underlying number of INM subpopulations in pollen.

The ice nucleation activity is in agreement with previous studies. From the first study on the ice nucleation activity of *B. pendula* (Diehl et al., 2001) to the discovery of the INMs (Pummer et al., 2012), the freezing temperature was reported between

-15°C and -18°C, which was also supported by other studies (Augustin et al., 2013; Burkart et al., 2021), some even showing the two distinct species at -15°C and -18°C (O'Sullivan et al., 2015; Dreischmeier et al., 2017). First, Dreischmeier et al. (2017) showed these highly efficient INMs in the supplement; however, they did not discuss them, and only later in Dreischmeier's Ph.D. thesis they were assigned to whole pollen grains rather than being INMs (Dreischmeier, 2019). More recently, *Class A* was mentioned in the context of cryopreservation in 100 µL volumes (Murray et al., 2022). Our study now

clearly shows the complete $N_m$ spectrum of *B. pendula* with three distinct INM classes with average freezing temperatures at -8.7°C, -15.4°C, and -17.4°C, which can be used to estimate the concentration of all INM classes. In addition, the fitting results (Fig. 1b) are in agreement with the assignment of the pollen INMs with the experimental data from Dreischmeier et al. (2017) and performed by de Almeida Ribeiro et al. (2023). However, these signals' robustness and physical nature require further investigation.



Combining the two ice nucleation assays has two crucial advantages: First, it allows the measurement of low-concentrated ice nuclei, and second, it will enable measurements down to about -35°C. The cooling rates vary significantly, namely 1 K min⁻¹ for the µL-assay and 10 K min⁻¹ for the pL-assay. However, previous studies showed that the cooling rate has a negligible effect on immersion freezing temperatures (Wright et al., 2013). Due to the low concentration, *Class A* INMs are only detectable with the µL-droplet assay. Both ice nucleation assays can measure *Class B* and *C* INMs. Moreover, the pL assay

allows us to conclude that no further classes of nucleators are present in the sample. When comparing the setups, we note that the difference in analyzed volume (when analyzing the same number of droplets) is a factor of $10^6$ (µL to pL). This results in a factor of $10^6$ higher sensitivity, which is visible in Figure 1a when comparing the start of the $N_m$ spectra of the respective assays. On the other hand, the background of our ultra-high quality (UHQ) water was between -21°C and -29°C for the µL-assay, a typical range for µL immersion mode assays (Miller et al., 2021) and (ii) between -32°C and -35°C when measured

with the pL-assay (see Figure S1). The background of the pL-assay is close to the homogeneous freezing temperature of cloud droplets at -38°C, allowing us to measure the full range relevant to atmospheric ice nucleation.

The combination of pL and nL droplets to obtain a complete ice nuclei spectrum of a sample has been suggested previously (O'Sullivan et al., 2014; Hader et al., 2014). In this study, we used pL and µL droplets and showed that pL and µL droplets (a factor of 1000 larger than nL) can also be combined, resulting in an even lower detection limit.

*B. pendula* INMs are often overlooked in terms of atmospheric relevance due to their low nucleation temperatures between -15°C and -18°C. Inorganic ice nuclei, e.g., mineral dust, are much more prevalent in this temperature range due to their high mass concentrations above the land surface. Naturally occurring ice nuclei with nucleation temperatures >-10°C are rare and dominated mainly by fungal and bacterial ice nuclei, which appear in low natural concentrations. We show that *B. pendula* pollen can contain highly efficient ice nuclei (*Class A* INMs) but at very low concentrations compared to the more common

*Class B* and *C* INMs. However, studies have shown that INMs found on *B. pendula* pollen are across the whole tree (Felgitsch et al., 2018; Seifried et al., 2020). Considering the wide distribution of *B. pendula* in the boreal forest, we speculate that *B. pendula* INMs might influence the total population of ice nuclei above -10°C near boreal forests. Understanding the emission, transmission, and nucleation mechanisms of ice nuclei active in this high-temperature regime is crucial because of their substantial influence on cloud micro-processes and subsequent impacts on weather and climate.

**3.2 Effects of freeze drying, freeze-thaw cycles, and heat on ice nucleation activity**

Upon establishing the presence of three distinct INM classes in *B. pendula* pollen, we further explored their stability and response to various conditions: (a) freeze-drying with subsequent hydration of the residue, (b) repeated freeze-thaw cycles, and (c) heat treatments.

Figure 2 shows how freeze-drying and subsequent hydration of the residue - while retaining the initial concentration of the

INM sample - drastically changes the $N_m$ spectrum. We observed an increase in the concentration of *Class A* INM, where $N_m$ increased by factors between 10 and 400 within six replicates of this experiment (five other measurements are shown in the supplement Figure S2). In contrast, Class B and C concentrations remained constant or decreased slightly by a factor of up to five.

To the best of our knowledge, the effect of freeze-drying has not been deliberately studied for biological ice nuclei. However,

previous studies have regularly freeze-dried samples before ice nucleation measurements (Zhang et al., 2016; O'Sullivan et al., 2015; Forbes et al., 2022; Schwidetzky et al., 2023; Rangel-Alvarado et al., 2015; Lukas et al., 2021). In particular, the widely studied Snomax® consists of freeze-dried fragments of *Pseudomonas syringae*. Unraveling how freeze-drying enhances the INM of *B. pendula* could deepen our understanding of the INM and its ice nucleation mechanism. In addition, studies using freeze-dried samples should prioritize the investigation of this effect.



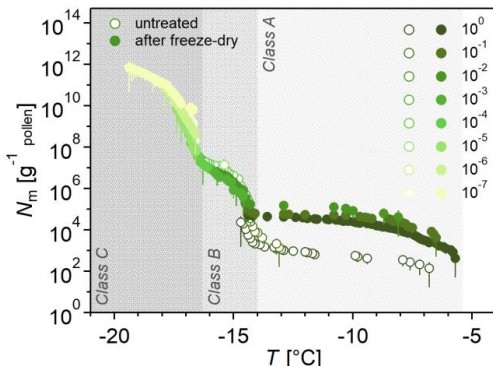


**Figure 2. Freeze-drying increases the ice nucleation activity of birch pollen washing water (BPWW).** Cumulative number $N_m(T)$ of INMs per gram of *Betula pendula* pollen after freeze-drying BPWW, dissolving in ultra-high quality (UHQ) water, and retaining the initial BPWW concentration. The freeze-dried sample (solid circles) increased in ice nucleation activity compared to the untreated sample (hollow circles). Ice nucleation activity consistently increased in the four independent experiments (see supplement), but the increase varied. The

legend on the left shows the dilutions of the sample, e.g., $10^{-1}$ corresponds to a 1:10 dilution.

Figure 3 shows that the $N_m$ spectrum remained stable throughout fourteen freeze-thaw cycles. However, at temperatures below -15°C, we observe a shift to higher concentrations of *Class B* INMs and an inverse effect on *Class C* INMs, both by a factor of ~5.

Experiments of this type have been performed with different biological INMs, resulting in significant shifts in the freezing

efficiency of bacterial ice nuclei (Polen et al., 2016) and much smaller effects on fungal and lichen INMs (Schwidetzky et al., 2023; Eufemio et al., 2023)

While the ice nucleation active component of bacteria and fungi are proteins (Green and Warren, 1985; Schwidetzky et al., 2023), it is unknown for lichens and *B. pendula*. The contrasting responses of biological ice nucleators to freeze-thaw cycles highlight the complexity of their ice nucleation mechanisms and underline the importance of further studies.

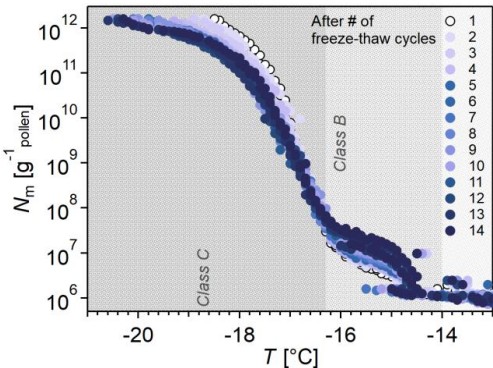


**Figure 3. Freeze-thaw cycles alter the ice nucleation activity of birch pollen washing water (BPWW) in lower-temperature regions. Cumulative number** $N_m(T)$ of INMs per gram pollen after fourteen freeze-thaw cycles. The color gradient indicates the number of freeze-thaw cycles, with the colors increasing in darkness with each cycle, as indicated in the legend on the right. The figure shows the temperature range where the repeated freeze-thaw cycles have affected the ice nucleation activity. Each cycle increases ice nucleation activity at -15°C

and decreases at -18°C.

Figure 4 shows the $N_m$ spectra of heat-treated samples. Heat treatment at 40°C for one hour had a minor effect on *Class A* INMs, decreasing the concentration by about ten, while *Classes B* and *C* were unaffected. A one-hour heat treatment at 78°C deactivates *Class A*, while *B* and *C* are slightly reduced. This effect is more pronounced when the temperature is increased to 98°C and even more pronounced when the heat treatment duration is increased from one to 24 hours. Heat treatment at 98°C

also increases ice nucleation activity to a slightly lower temperature. Another way of looking at these results could be a



complete shift of the $N_m$ spectra downwards with increasing treatment temperature and duration to a hypothetical line at the detection limit at ~$10^2$ g$^{-1}$.

Heat treatments serve multiple purposes. Firstly, they are often used on field samples to determine whether the ice nuclei are of biological origin, as many biological molecules, particularly proteins, are heat-sensitive and can denature at high

temperatures, losing their activity. Secondly, these treatments provide insights into the stability of ice nuclei, which is crucial for understanding their nucleating mechanisms and inferring their chemical composition.

Previous studies on the thermal stability of birch INMs showed mixed results. Pummer et al. (2012) reported no change in ice nucleation activity after dry heating up to 111°C for 1 h, but a complete loss of activity after heating to 202°C for 1 h (Pummer et al., 2012), but in this work the sample was dried before the heat treatment. Daily et al. (2022) also reported a loss of activity

after dry heating to 250°C for 4 h and no loss after a wet heat treatment at 100°C for 30 min. Dreischmeier et al. (2017) also used a wet heat treatment (as in our study) and showed a concentration-dependent effect. Heat treatment of 90°C for 3 h resulted in a complete loss of activity in a less concentrated sample but had no impact on a thousandfold more concentrated sample (Dreischmeier, 2019). This is consistent with our results from a dilution series, for which higher treatment temperature and duration, fewer dilutions had to be measured until the sample was homogeneously frozen. Dreischmeier (2019) and

Pummer et al. (2012) used pL-assays, which did not allow them to measure *Class A* INMs. The measurement of a dilution series allowed us to obtain a more comprehensive understanding of the thermal degradation process.

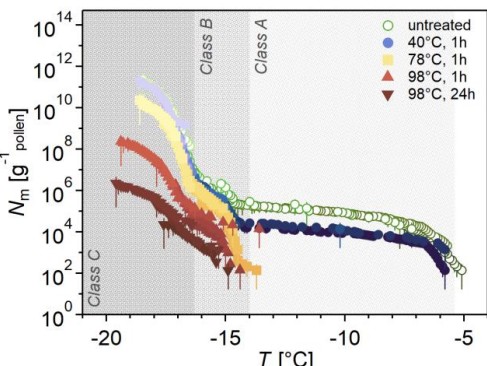

**Figure 4. Heat treatment diminishes ice nucleation activity in birch pollen washing water (BPWW), especially at T > -15°C.** Cumulative number $N_m(T)$ of INMs per gram *Betula pendula* pollen. INMs were ice-affinity purified and then heated (40°C – blue circles,

78°C – yellow squares, 98°C – red triangles) for 1 h (also 24 h for 98°C, downward triangles), the color gradients indicate dilutions. Heat effects depend on treatment temperature, duration, and freezing temperature. Ice nucleation activity > -temperatures of 40°C reduce 15°C, while ice nucleation activity < -15°C is only altered at temperatures of 98°C.

To summarize, we show how ice nucleation activity of birch INMs can be influenced: (i) freeze-drying increases *Class A* INM concentration, (ii) freeze-thaw cycles increase *Class B* and decrease *Class C* INM concentration, and (iii) heat treatments

decrease INM concentration.

This section shows how subtle changes in sample treatment can significantly alter the ice nucleation activity of birch INMs. While heat treatment is intended to change the activity of a sample, repeated freezing or freeze-drying are usually (un)intended steps for or during sample preparation that can significantly impact the results. We suggest that in further studies, the steps of sample preparation, both in the lab and field studies, are documented in detail. Understanding how ice nucleation activity is

affected can provide important insight into the stability and mechanisms of ice nuclei.

Recently, Murray et al. (2022) proposed that highly efficient birch INMs are likely to be large aggregates of smaller subunits. This aligns with our hypothesis that *Class A* INMs are aggregates of *Class B* and *C* INMs, only differing in size. We propose that *Class C* represents the smallest aggregates or even single subunits, *Class B* consists of small aggregates, and *Class A* comprises significantly larger aggregates that remain viable only under specific conditions (*Class A > Class B > Class C*). In

that sense, freeze-drying provides the condition for aggregation of *Class A* INMs, while freeze-thaw cycles only induce smaller



aggregation of *Class B* INMs. The exposure of INMs to heat leads to the deactivation or disassembly of the most fragile *Class A* INMs. In contrast, *Class C* INMs – small aggregates or single subunits – are stable and can withstand elevated temperatures for extended periods. To prove our hypothesis and investigate how aggregation could be a pivotal process for birch INMs, we conducted size-selective filtration and DLS measurements focusing on the size of the INM. We used CD, fluorescence, and

IR spectroscopy to investigate the chemistry of the INMs.

### 3.3 Investigations on INM Aggregation

We analyzed the size of the INMs by (a) using size-selective filters and subsequently measuring ice nucleation activity and (b) measuring particle sizes by dynamic light scattering (DLS) of the heat-treated samples, which differed in INM concentration and freezing temperatures. Figure 5 shows the $N_m$ spectra of the filtrates from one filtration series. Firstly, the ice nucleation

activity in terms of $N_m$ and freezing temperature decreased with decreasing pore sizes. All INM classes passed through the 300 kDa filter. The 100 kDa filter filtered *Class A* INMs, while *Class B* INMs were filtered by the 10 kDa. *Class C* passed through all filters. This experiment shows that the INM classes differ in size and that the larger the INMs, the higher the freezing temperature.

The unfiltered sample had a low concentration of *Class A* INMs. However, this trend is derived from five independent

experiments (see supplement Figure S5), but only one experiment is shown in Figure 5 for simplicity. With each filtration, the overall concentration decreases (except for 300 kDa), which we believe is due to a loss of INMs in the filter membrane. The < 50 and < 30 kDa fractions show similar concentrations in all replicate measurements, and thus, the cutoff ranges are not differentiable with our µL-droplet assay. We also note that individual droplets in the < 50 kDa fractions froze above -15°C due to this experiment's uncertainty.

MWCO filters, typically used for protein concentration, have nominal cutoff masses that rely on molecular density and shape assumptions. Therefore, we refrain from drawing direct conclusions about the INM mass or size and instead focus on the observed size trend.

Previous studies by Pummer et al. (2012) and Dreischmeier et al. (2017) employed MWCO filtrations and concluded that the INMs must be > 100 kDa, which is directly contradicted by our results. However, it is essential to note that both studies used

pL-assays, which, as we have previously shown, have a lower sensitivity than the µL-assay we used here. Hence, we conclude birch INMs are smaller than previously assumed, raising questions about their nucleation mechanism. However, the nominal MWCO filter sizes should not be regarded as absolute measures of INM size but can serve as indicators of the size range. In 2015, Pummer et al. (2015) compared various biological INMs to a parameterization of the critical ice embryo size, crucial for ice nucleation, over nucleation temperature (Zobrist et al., 2007) and argued that birch INMs, due to their size, nucleate ice at

low temperatures (-15°C and -18°C) due to their size. Our results are only partially consistent with this study. Depending on the size of the three INM classes we describe, an accurate size measurement is needed to prove or disprove this hypothesis, and MWCO filters are inadequate.




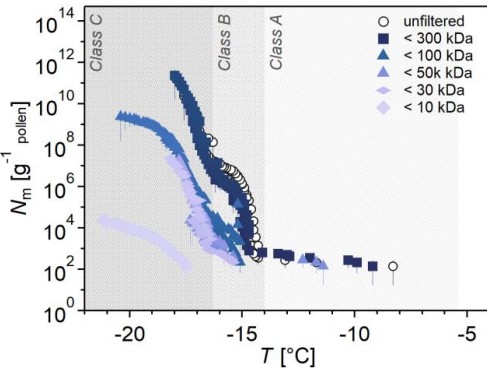

**Figure 5. Investigation of INM sizes by size-selective filtrates of birch pollen washing water (BPWW).** (a) Cumulative number $N_m(T)$ of INMs per gram of *Betula pendula* pollen. BPWW was size-selectively filtered with nominal filter sizes of 300, 100, 50, 30, and 10 kDa. The ice nucleation activity decreases with smaller filter sizes in concentration and initial freezing temperature. The colors increase in brightness with smaller cutoff sizes. Four independent experiments were performed (see supplement). Unfiltered sample refers to BPWW, which, during preparation, is filtered with a 220 nm filter.

Figure 6 shows the hydrodynamic radius ($R_h$) of ice-affinity purified BPWW before and after heat treatment as measured by DLS. The radii range from 240°C (untreated) to ~280 nm (40°C) down to ~115 nm (98°C, 1 h). The higher the heat treatment temperature, the smaller the radii. The untreated sample has a similar radius to the sample treated at 40°C but a much larger uncertainty (calculated from 4 independent samples).

DLS gives only a single particle size, while it is highly likely that there are different particle sizes within the sample. We are observing the largest particles as the intensity of light scattering is disproportionately affected by larger particles.

Figures 5 and 6 clearly show larger INMs have higher freezing temperatures, strengthening the aggregation hypothesis. The current idea is that aggregation is essential for ice-nucleating molecules and has been discussed in the context of ice-binding proteins (IBPs) from bacteria and insects (Burke and Lindow, 1990; Govindarajan and Lindow, 1988; Gurian-Sherman and Lindow, 1995; Kozloff et al., 1991). These proteins encompass antifreeze proteins (AFPs) and ice-nucleating proteins, each playing distinct roles in ice formation (DeVries and Wohlschlag, 1969; Bar Dolev et al., 2016). IBPs are characterized by motifs such as anchored clathrates and ice-like structures, typically found in β-sheet formations (Liou et al., 2000; Leinala et al., 2002; Graether and Jia, 2001; Hakim et al., 2013). These motifs confer a high affinity for ice. AFPs utilize these motifs to inhibit ice growth by binding to ice crystals (Raymond and DeVries, 1977). In contrast, INPs, primarily studied in bacterial contexts, form larger aggregates (Ling et al., 2018; Burke and Lindow, 1990). These aggregates enhance the proteins' ice-nucleating capabilities, with larger aggregates correlating with higher nucleation temperatures (Qiu et al., 2019; Schwidetzky et al., 2023). The aggregation in bacterial ice nuclei is facilitated by specific protein sequences that enable aggregation (Burke and Lindow, 1990; Gurian-Sherman and Lindow, 1995; Lukas et al., 2022; Govindarajan and Lindow, 1988). A similar idea is proposed for fungal ice nuclei, which are also of proteinaceous nature (Schwidetzky et al., 2023), for birch INMs aggregation has initially been suggested by Dreischmeier et al. (2017) and Murray et al. (2022). However, the exact molecular composition of birch INMs remains elusive. Hence, caution is necessary when adapting the aggregation theory from bacterial or fungal ice nuclei. This uncertainty underscores the importance of chemical analysis.



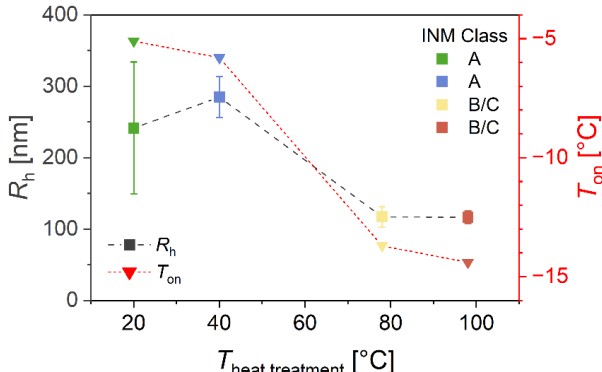

**Figure 6. INM sizes determined by dynamic light scattering (DLS) of heat treated BPWW compared to freezing onset temperature $T_{on}$.** The hydrodynamic radius $R_h$ (nm) of particles in ice-affinity purified BPWW, measured by DLS, shows a decline in the size of the largest particles with increasing heat treatment temperature. Along with it the freezing onset temperature of the undiluted sample decreased (freezing onset temperatures were determined from Fig. 4). Error bars were calculated using error propagation combining the measurement errors from multiple measurements. Untreated BPWW was plotted as heat treated at 20°C for practical purposes. Replication of the 40°C heat treatment was not possible, and only measurement has been acquired. Therefore, we assumed an error of 10%.

### 3.4 Chemical analysis of birch INMs

Initially, we employed fluorescence and infrared spectroscopy to analyze ice affinity purified BPWW, methods previously used by Pummer et al. (2013), Dreischmeier et al. (2017), and Burkart et al. (2021). First, fluorescence spectroscopy showed signals typical for aromatic amino acids, proving the presence of proteins (see Figure S6). Second, infrared spectroscopy showed typical protein bands (amide I, II, III bands) and typical polysaccharide bands (sugar skeletal vibration and COC glycosidic vibration); the entire spectrum can be seen in the supplement Figure S7. These findings are consistent with existing literature and show proteinaceous and polysaccharidic signals. We further analyzed BPWW using circular dichroism (CD) spectroscopy.

Figure 7 shows the CD spectra of ice-affinity purified BPWW before and after heat treatment. The maximum is at ~185 nm and the minimum at about 210 nm, and the signal increases afterward until 250 nm. The signal of the heat-treated sample differs between 185 and 210 nm. With the CD spectrum of BPWW, we used the BeStSel tool to estimate the ratios of secondary structures to each other based on the CD spectra. The database comparison shows BPWW has about 30% beta-sheets and 5% alpha-helix structures. This is noteworthy since many of the highest active (in terms of freezing temperature) biological ice-nucleating proteins have considerable beta-sheet structures (Lukas et al., 2021; Garnham et al., 2011; Kajava and Lindow, 1993). However, we know several proteins in our sample, which are not ice nucleation active such as the *Bet v* allergens (citation?).

Birch INMs were initially suggested to be composed of polysaccharides based on their high-temperature stability and various chemical and enzymatic experiments (Pummer et al., 2012, 2015; Dreischmeier et al., 2017). In contrast, studies focusing on specifically denaturing proteins and fluorescence spectroscopy suggest that proteins are at least involved in the ice nucleation activity of birch INMs (Burkart et al., 2021; Tong et al., 2015; Felgitsch, 2019). Burkart et al. (2021) also speculate that highly glycosylated proteins might be the ice nucleation active component. While our findings add weight to the theory of proteinaceous involvement in birch INMs, they also highlight the inherent complexity in deciphering the exact composition of birch INMs. This uncertainty presents a challenge for future research.



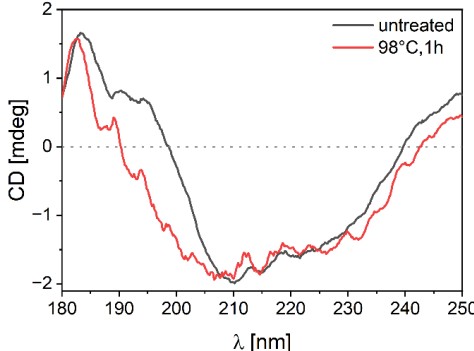

**Figure 7. Circular dichroism spectrum of birch pollen washing water (BPWW), untreated and heat treated.** The spectrum shows the signal of ice-affinity purified BPWW untreated (black) and heat-treated (red), with a minimum molar ellipticity at ~ 205 nm and a local maximum at ~ 195 nm. The heat-treated signal differs mainly in the 185 to 210 nm region, indicating a structural change.

## 4. Conclusions

Our study re-examines the INMs of *Betula pendula* pollen, shedding new light on their complex ice nucleation capabilities. We identified three classes of INMs with average freezing temperatures of -8.7°C (*Class A*), -15.4°C (*Class B*), and -17.4°C (*Class C*). The concentrations from *Class A* to *C* increase by multiple orders of magnitude. Combining two ice nucleation assays with different droplet sizes (3 µL – TINA and pL-droplets – VODCA) allowed us to measure with high sensitivity and at temperatures down to -34°C.

We also evaluated physical influences on the ice nucleation activity of our samples. First, we found that freeze-drying increases the concentration of *Class A* INMs up to factors of four hundred, while the concentration of *Classes B* and *C* remains the same or decreases slightly. Second, we found that repeated freeze-thaw cycles led to an increase in *Class B* and a decrease in *Class C* concentrations, both by a factor of five, while *Class A* was unaffected. Third, we showed how heat reduces the concentration of all INMs. *Class A* INMs were completely inactivated after 1 h at 78°C. *Class B* and *C* INMs decreased after 1h at 98°C, and after 24 h at 98°C, only *Class C* INMs remained. This demonstrates how subtle (unintended) differences in sample treatment can significantly alter the ice nucleation activity of birch INMs.

At last, we focused on investigating the size and chemistry. Size-selective filtrations showed that the smaller the INMs, the lower the freezing temperature. DLS measurements of untreated and heat-treated INMs show a decrease in size with increasing heat treatment temperature. Chemical analysis, including CD-, fluorescence- and infrared spectroscopy, show proteinaceous and polysaccharidic signals.

Most previous studies on birch INMs have only reported activity at -15°C and -18°C (Diehl et al., 2001; Pummer et al., 2012; O'Sullivan et al., 2015; Dreischmeier et al., 2017; Augustin et al., 2013), with one study reporting up to -6°C (Murray et al., 2022). Our results confirm that INMs from *B. pendula* can induce heterogeneous freezing at temperatures at high subzero temperatures (up to -5.4°C), and we can estimate the concentration of these highly efficient INMs. Furthermore, it allows field measurements to search for birch INMs in the atmosphere actively. Tree-based INMs are often overlooked as a source of atmospherically relevant ice nuclei of biological origin. However, studies on *B. pendula* and *Pinus sylvestris*, another tree species with pollen that contains INMs, have shown an abundance of INMs on trees' surfaces (Felgitsch et al., 2018; Seifried et al., 2020, 2023). These tree species are widespread in the boreal forest, making them an important reservoir of INMs, and could potentially be large emitters of INMs.

Considering all our results, we propose that *B. pendula* INM classes different-sized aggregates of small subunits. The larger the aggregate, the higher the freezing temperature. *Class C* INMs are the smallest aggregates or single units. A certain stable



aggregate size of the subunits forms the *Class B* INMs, and *Class A* INMs are even larger. *Class A* INMs require specific

prerequisites to form, and our results suggest that freeze-drying can induce this process. At the same time, repeated freeze-
thaw cycles seem to induce only minor aggregation of *Class B*. Heat leads to disassembly of the aggregates, and our results
suggest that the larger the aggregates, the lower the temperature required for disassembly. Nonetheless, our results open further
questions: a) about the sizes of aggregates, b) the chemical composition of their subunits, and c) how and why aggregates
foster ice nucleation. Up to our understanding, there are four central physical and chemical reasons for enhanced ice nucleation

activity related to aggregates:

    i)    Water activity is a thermodynamic predictor of kinetic ice nucleation. It reveals a direct relationship to the water
          structure at the INM surface, allowing interpretation of the interaction between INM and critical ice nuclei. (Knopf
          and Alpert, 2023)

    ii)   Water in nano-sized confined geometries at the surface of INM aggregates can serve as the origin of ice formation,

leading to stacking disordered ice (Nandy et al., 2023; Knopf and Alpert, 2023)

    iii)  Amino acid *TxT* motifs, where *T* is threonine, and *x* is a non-conserved amino acid, can order water molecules into a
          stacking disordered ice structure attributed to chemical functional groups capable of hydrogen bonding to water
          molecules composed of hydrophilic-hydrophobic patterns.  (Alsante et al., 2024; Qiu et al., 2019)

    iv)   The ice nucleation process releases latent heat, which is the kinetic energy of the water molecules of the liquid

transformed into infrared radiation. Aggregates exhibit an enhanced number of degrees of freedom, allowing a more
          efficient uptake of kinetic energy and a broader emission of IR radiation.

Elucidating the chemistry and physics of the INM will help further understand how they nucleate ice. (Felgitsch et al., 2018;
Seifried et al., 2020, 2023)In summary, our research contributes significant new insights into the nature and behavior of *B.
pendula* INMs and underscores their potential as critical players in atmospheric ice nucleation processes. This understanding

is crucial for refining weather and climate models, where the role of biological ice nucleators has been historically
underappreciated.



**Data availability**

All data are available from the corresponding author upon request.

**Author contributions**

HG, JFN, KM and TMS conceptualized the project. HG, JFN and KM supervised FR. FR performed the experiments with support of NB and RS. All authors discussed the results. FR prepared the manuscript with input and revisions from all co-authors.

**Competing interest**

The authors declare no conflict of interest.

**Acknowledgements**

The authors would like to thank Ellen H.G. Backus, Clara M. Saak and Friederike Strahl from the Department of Physical Chemistry of the University of Vienna for valuable discussions and input. The authors also thank Rosemary Eufemio from Boise State University for her support with the HUB-backward fitting of the ice nucleation data.

**Financial support**

The authors acknowledge TU Wien Bibliothek for financial support through its Open Access Funding Programme. We gratefully acknowledge support by the FFG (Austrian Research Promotion Agency) for funding as part of project no. 888109 (Lab on a Drone).



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
