# Peer review of "Aggregation of ice-nucleating macromolecules from *Betula pendula* pollen determines ice nucleation efficiency."

_EGUsphere, 2024_

## Referee Comment (RC2)

Comments to "Aggregation of ice-nucleating macromolecules from Betula pendula pollen determines ice nucleation efficiency" by Florian Wieland et al. (Manuscript ID: egusphere-2024-752).

Based on the twin-plate ice nucleation assay (TINA) and Vienna optical droplet crystallization analyzer (VODCA), the authors explored the ice nucleation activity of ice nucleating macromolecules (INMs) in Betula pendula pollens. The authors found three different types of INMs that caused freezing at -8.7°C, -15.7°C, and -17.4°C, respectively. The authors also found freeze drying, freeze-thaw cycles, and heat treat can alter the ice nucleation activity, which was attributed to size-varying aggregates. Overall, the science is interesting, and manuscript is written well. If the following comments can be fully addressed, the manuscript may be publishable in *Atmos Chem Phys.*

Major comments:

1. The droplets in TINA experiment are shaped by the well. However, the droplets in VODCA are shaped by surrounding aqueous phase. How the different morphology and contact angles of generated droplets will influence the measured ice nucleating activity?

2. The authors treated the BPWW at 40 °C, 78 °C and 98 °C for 1 h or 24 h respectively. What is the relevance of such high temperatures to ambient air especially the troposphere air?

3. Why the error bars of hydrodynamic radius $R_h$ of untreated BPWW are significantly higher than other samples (Figure 6)?

4. Figure 1: in the left panel, the data points for pL droplets scattered significantly. One temperature corresponds to several $N_m$ values, how to explain this?

5. Line 127: is the comment 'VODCA is optimal for high INM concentrations at low temperatures, while TINA is better for low INM concentrations at higher temperatures' reasonable?

6. Figure 4: it looks like the drying time duration significantly influence the $N_m$. Whether the chemical composition has been changed? FTIR data is preferred to confirm this point.

7. Figure 5: what is the relative fraction change of different size INMs for untreated and treated BPWW samples?

---

## Author Comment (AC1)

**Referee comment 1**

Reviewer comment by Gabor Vali on "Aggregation of ice-nucleating macromolecules from *Betula pendula* pollen determines ice nucleation efficiency." by Florian Wieland, Nadine Bothen, Ralph Schwidetzky, Teresa M. Seifried, Paul Bieber, Ulrich Pöschl, Konrad Meister, Mischa Bonn, Janine Fröhlich-Nowoisky and Hinrich Grothe.

The paper presents a rather extensive set of measurements and interpretations aimed at obtaining better understanding of the ice nucleating ability of birch pollen. The topic is of interest both from the point of view of basic nucleation concepts and as a potential contribution of the pollen to atmospheric cloud processes.

The ice nucleating ability of macromolecules from birch pollen has been the subject of various prior publications over the past two decades with a focus on freezing temperatures near -15°C. Some activity was found at higher temperatures as well and that diversity prompted studies to identify its causes. That's the main focus of the current paper. Activity at temperatures around - 9°C is shown more definitely than previous work. The authors conclude that aggregates of macromolecules may be responsible for this.

**Response:** We thank the referee for carefully assessing our work and all thoughtful comments and will answer each comment separately below.

Action Taken: Our comments are in blue, while the referee comments are in black. Parts of the manuscript are in orange, with changes and additions underlined.

**[Referee 1 – Comment 1]** The main feature of the empirical work is the use of two different instruments with differing sample volumes and thus obtain freezing nucleus spectra over a broad temperature range. While data over this large temperature range could also have been obtained by successive dilutions of the sample and fusion only the instrument with larger sample volumes, the use of two devices was no doubt a practical help, and the agreement in the results from the two techniques adds further confidence to the results. The authors overstate to some extent that the use of two devices was crucial.

**Response:** We thank the referee for his insightful comment. The biggest disadvantage of the  $\mu$ L-assay is the much warmer background water freezing, which makes it experimentally challenging to obtain data below -25°C. In contrast, the pL-droplet assay has a background freezing of -34°C, and we do not know of any assay with  $\mu$ L-droplets that can reliably measure down to this temperature. Hence, while we agree with the referee that successive dilution is the preferred choice to cover a broad range of concentrations, using two assays offers additional advantages. Since we do not know if anything nucleates below -25°C without using the pL assay, so, the pL assay allows us to exclude the presence of nucleating substances presented below -25°C (e.g. an additional nucleation step could have been present at -28°C but we would not have detected it with using the  $\mu$ L assay only.)

Action Taken: Based on the referee's recommendation, we shortened the argument in the results section and moved most of it to the method section, which is a better fit.

In the method section, we provide this argument, which is slightly adapted from the original manuscript:

"TINA analyzes 98 droplets of 3 µL each and up to 8 samples simultaneously. In contrast, VODCA measures around 100 droplets ranging from 1.8 pL to 33.5 pL. As pL droplets are six orders of magnitude smaller than µL droplets, the probability of homogeneous and heterogeneous freezing events is much higher in µL droplets. Classical nucleation theory suggests smaller droplets, such as those in VODCA, freeze at lower temperatures. The background freezing temperature in VODCA is around -33°C, whereas with TINA, the background freezing of UHQ water has been measured between -20°C and -30°C. In addition, larger droplets contain more INMs than small droplets. Therefore, the probability of ice nucleation by high-temperature active INMs (present only at low concentration) is increased in large droplets, as the most efficient INM nucleates the whole droplet. Consequently, VODCA and TINA can be considered complementary techniques: While TINA is better for low INM concentrations, VODCA allows measuring characteristic freezing temperatures down to -34°C. "

**We have kept the following argument in the results section:**

"The combination of two ice nucleation assays is beneficial for two reasons: (1) low-concentrated ice nuclei can be detected, and (2) measurement down to about -34°C. Due to the low concentration, *Class A* INMs are only detectable with the  $\mu$ L-droplet assay, while the pL-assay allows us to conclude that no further classes of nucleators are present in the sample."

**[Referee 1 – Comment 2]** As mentioned above, the results of the freezing assays are reassuringly solid. The spectra shown have relatively little scatter. While the authors indicate (lines183-188 and in the Supplement) that variations results with time, sample origin and other factors, they do not indicate what conditions were in place for the results shown in the different graphs.

**Response:** We thank the referee for his positive assessment of our data quality and for pointing out that experimental details are missing. We replicated measurements a year after the original experiments and tested the influence of storage conditions. Most of the data presented is from our original experiments. Data from "Pollen A0 and B0" are from the original experiments, while data from "Pollen A1/A2 and B1/2" are from experiments replicated a year after.

Action Taken: We have added details of the experiments to the manuscript and supplement.

**[Referee 1 – Comment 3]** The basis for discussing multimodal freezing activity is the presence of three areas of steep slopes in the observed cumulative spectra, resolved into three components of the differential spectrum via the HUB method (line 196). The advantages and disadvantages of this approach, compared to direct calculation of the differential spectrum from observed freezing temperatures of individual drops are debatable, but are not essential for the present topic. The intermediate leveling of the spectrum is clear evidence for separating activity into different temperature regions. The authors call this A, B and C class nucleating agents. By subjective examination of Fig 1, classes A and C seem to have stronger evidence than B. Uncertainties attached to the resolved spectra are not specified in the paper.

**Response:** INMs of *Class B* appear in all our results as a shoulder signal to *Class C* but are not as pronounced in the spectra as *Class A* and *C*, as the referee pointed out. Nevertheless, all measurements of the cumulative spectra (except those deactivated by heat treatment and filtered out by filtration) show the *Class B* signal very reproducibly. Dreischmeier et al. (2017) and O'Sullivan et al. (2015) have also found these two types of IN - which we refer to as *Class B* and *C*. In particular, the differentiated spectrum supports our argument. We have chosen the HUB

method to plot the differential spectrum because it uses a fit of the cumulative data, which averages over the dilutions. We, therefore, believe that one should not look at individual measurements only but at a series of measurements for which one can give a specific uncertainty. The differential spectrum shown in Fig. 1b and the reproducibility of the signal in all other (relevant) measurements is robust enough evidence, in our opinion.

**Action Taken: We added this statement to section 3.1.**

"Class B INMs are arguably hard to distinguish from Class C INMs, as their activity overlaps. However, the distinct rise at ~-15°C (Class B), prior to the rise at ~-19°C (Class C) in the cumulative spectrum is very reproducible in all our measurements, suggesting that Class B is not a measurement artifact but indeed a separate Class of INMs. "

[Referee 1 – Comment 4] The principal focus of the paper, as indicated by the title and the explanations offered in the Conclusions is the notion that aggregates of smaller units are responsible for activity found at temperatures higher than about -15°C. Why that process would lead to activity at distinct temperatures (spectral peaks) is not discussed. To this reviewer, that question is so fundamental. Perhaps the preservation of the spectral shape (Fig. 5) is the main justification for discrete sizes and discrete regions of activity. If so, that point should be made emphatically, beyond the statement on line 356. Better yet, the point should be made explicit by plotting differential spectra for the samples with different filter cutoffs. However, the sample to sample variations and the small number of freezing events for Class A probably make such analysis impractical. Therefore, the issue seems to have no clear answer in the data so far obtained. Perhaps the authors do have other arguments for why there are preferred sizes or shapes of aggregates which correlate with the preferred nucleation temperatures; if so the readers need to find it in the paper. Neither of the four possible explanations offers clarity on this. Neither does the widely used assumption that larger sizes of a nucleating substrate offer greater chances of finding sites of given activity (a theory not entertained by the authors) explain the discreteness observed. That notion, coupled with assumed frequency distributions of sizes would explain the results for any one peak but is hard to separate from the effects of random size fluctuations of embryos.

**Response:** Indeed, we hypothesize that aggregation is responsible for the three different freezing temperatures. Larger aggregates lead to higher ice nucleation temperatures, which we derive from our data in Figure 5 and 6. Plotting the differential spectra, in our opinion, does not better represent this argument since, as the referee suggests, the sample-to-sample variations make this analysis rather impractical. We believe that presenting our data in this way, combined with the data in the supplement, is the best way to convey this message. But as the referee suggests we will make this statement more explicit in the revised manuscript.

In addition, we describe that different nucleation mechanisms could be responsible as the size of the aggregate increases, this is indeed a speculative hypothesis that we try to tackle in the conclusion. However, the referee is right that all this is based on a thin layer of experimental data. Therefore, we will more precisely separate the experimental facts from the interpretations and the hypotheses.

**Action Taken:** We restructured section 3.3 and explained our results of the filtration and DLS experiments more explicitly, as the referee suggested. Furthermore, we moved the more speculative interpretation into explaining our hypothesis in the conclusion section.

**We rephrased the explanation of the filtration experiments:**

"We conducted five independent filtration experiment series of untreated BPWW. Figure 5 shows the  $N_m$  spectra of the filtrates from one filtration series, for simplicity, the full dataset is shown in Figure S5. Generally, all samples showed low concentrations of *Class A* INMs, we can still see that the 300 kDa filter does not affect the sample, whereas the 100 kDa removes all of *Class A* INMs and decreases the concentration of *Class B* and *C* INMs. The < 50 and < 30 kDa fractions almost show no *Class B* INMs. Individual droplets in the < 50 kDa fractions froze above -15°C due to this experiment's uncertainty. The <10 kDa fraction shows almost exclusively *Class C* INMs, with single freezing events at higher temperatures. Hence, we can confidently say the ice nucleation activity in terms of  $N_m$  and freezing temperature decreased with decreasing pore sizes. This suggests the three INM classes are different in terms of size, with *Class A* being the largest and *Class C* being the smallest aggregate."

**We rephrased the explanation of the DLS measurements:**

"Figure 6 shows the hydrodynamic radius ( $R_h$ ) of ice-affinity purified (hence freeze-dried) BPWW before and after heat treatment as measured by DLS. The radii range from 240 nm (non-heat-treated) to ~280 nm (40°C) down to ~115 nm (98°C, 1 h). The radii are correlated to the freezing onset temperature as measured with the µL-assay and shown in Figure 4. We can see that the non-heat-treated and 40°C treated samples have larger radii of ~250 nm and higher freezing onset temperatures of around ~-6°C. In contrast, the 78°C and 98°C treated samples have smaller radii of ~180 nm and lower freezing onset temperatures of around ~-15°C. From the freezing spectra in Figure 4 we can see, that while the non-heat-treated and 40°C treated sample contain *Class A* INMs, the 78°C and 98°C, only contain small concentrations of *Class B* and mostly *Class C* INMs. The data evidently shows that the higher the heat treatment temperature, the smaller the radii, directly correlating to lower freezing onset temperatures and INM classes. "

[Referee 1 – Comment 5] Results presented on the effects of freeze-drying, heat treatment, and of freeze-thaw cycles are good material for assembling evidence on the chemical composition of the pollen INMs, but do not constitute significant additions to the main point of paper.

**Response:** Our intention was not to go into the chemical composition here. Instead, the data in section 3.2 is intended to show how the three IN classes (especially Class A) are influenced by different conditions. These experiments show that the three INM classes react differently to the exposed conditions. We consider this extremely important as it supports our argument that the nucleation mechanism for the three classes might be different, e.g. classes B and C are more robust to heat than Class A, while Class A is more uplifted by freeze-drying but not affected by freeze-thaw cycles. As far as we know, this is the first example in which such effects have been demonstrated.

Action Taken: This point was formulated more clearly in the revised manuscript to address this comment. Here is one example of how we rephrased.

"Although heat treatments at high temperatures are arguably not relevant for ambient atmospheric processes, they provide valuable insight: Firstly, these treatments reveal a diverging response of the INM classes to heat, which raises further questions about possibly varying chemistry, which could also mean that the nucleation mechanisms are different. Secondly, they are often used on field samples to determine whether the ice nuclei are of biological origin, as many biological molecules, particularly proteins, are heat-sensitive and can denature at high

temperatures, losing their activity. Our data suggests a simple heat treatment is insufficient to classify whether ice nuclei are of biological origin or not."

[Referee 1 – Comment 6] A comment on writing style: Section 3 is titles "Results and discussion" and the content follows that double intention. The mixed presentation of measurement results, and discussions of their possible implications can be helpful in digesting the significance of data, but it is carried a little too far in this paper. Conclusions are pre-announced, references to other work are not restricted to facts but also to speculations, and in general much is included in the Results section that should be left to the Conclusions. The authors might want to re-examine this.

**Action Taken:** To address this, we moved speculations to the conclusion and tried to avoid too strong interpretations in the results & discussion section.

**[Referee 1 – Comment 7]**

Minor comments:

Key figure: this illustration is overly simplistic in that aggregation in the atmosphere is not likely to result in units of the same species. Many different components of the atmosphere are likely to participate in forming any aggregates. Coagulation, in general, is driven by forces that do not distinguish one substance over another or does s to a very minor degree.

**Response:** We agree that the key figure suggested aggregation occurs during atmospheric transfer, which we did not intend.

Action Taken: We have adapted the key figure: left - initial version, right - revised figure

Line 39: location and time of year drive many of the factors mentioned.

**Response:** We have added the following in line 39:

Action Taken: "... data on INPs, such as composition, concentration, and initial freezing temperatures – many of these factors strongly depend on location and time of year – is crucial to understanding ..."

Line 43: nucleation is indicated to be **at** -15°C. Usually there is a spread in temperatures rather than a fixe value.

**Response:** We have rephrased the sentence to:

Action Taken: "...can induce heterogeneous ice nucleation around -15°C..."

Line 58: ".. in the supplement ..." is meant to refer to the publication by Dreischmeier et al.

**Response:** We changed the phrasing:

Action Taken: "...also published results in the publications' supplement showing ... "

Line 66: Unclear whether the authors here mean INP or INM.

Response: It should be INM.

Action Taken: "...(i) the different subpopulations of pollen INM and their initial freezing temperatures, ..."

Line 68: "Therefore " hardly follows from the preceding sentence.

**Response:** We removed *Therefore*.

Action Taken: "We used two complementary ice nucleation assays ..."

Lines 79-84: For readers not familiar with the process, it might be helpful to clarify how the sample preparation leads to macromolecular ice nucleators, not pollen grains or particles. "Extraction time" is not explained.

**Response:** The section has been rephrased.

**Action Taken:**

"The pollen grains were extracted for 6 h, during which the mixture..."

"..., and the supernatant was filtered through a 0.2 µm syringe filter (VWR, Radnor, PA, USA) to remove any pollen grains and their fragments from the sample. The obtained sample (BPWW) now only contains (macro-)molecular components."

Line 92: RT presumably means room temperature. Not a generally known acronym.

**Response:** We removed the acronym since it is not used anywhere else in the manuscript.

Action Taken: "...the samples were cooled to room temperature and analyzed..."

line 117: A correction factor for cooling rate effect is reasonably well known: about 2°C for a factor ten in cooling rate. This correction could be applied rather than dismissing the influence altogether.

Response: We added this information to the according statement in the methods section.

Action Taken: "The cooling rates are very different in 1 K min-1 for TINA and 10 K min-1 for VODCA. Previous studies showed that the cooling rate could influence the freezing temperature (Vali, 2014; Wright et al., 2013). One study suggests a factor ten in cooling rate can approximately shift freezing temperatures by up to ~2°C (Vali, 2014), in our case this means the pL-assay data might be shifted to lower temperatures, due to the higher cooling rate, which is not accounted for in our data, since an intercomparison of various ice nucleation assays showed good agreement of our pL-assay with other ice nucleation assays (DeMott et al., 2018)."

Sections 2.2.2 to 2.2.5: most readers would benefit from some explanation of what is expected to be learned from these measurements.

**Response:** We agree and added additional information in the method section.

Action Taken: We added these statements about the methods used.

"Dynamic light scattering (DLS) was measured to obtain information about the hydrodynamic radius of particles suspended in our sample. Particle sizes are measured with a Nicomp 380..."

"Circular dichroism (CD) determines information about chiral molecules in a sample, often used to investigate secondary structures of proteins found in BPWW. CD spectra were recorded..."

"Fluorescence spectroscopy detects the autofluorescence of BPWW, which can be assigned to various biological molecules, such as various proteins containing aromatic amino acids. Fluorescence excitation-emission\_maps (FEEM) were..."

"Vibrational spectroscopy, including infrared spectroscopy, records inter- and intramolecular vibrations, leading to chemical information such as functional groups. Thus, the sort of biomolecules in our samples, e.g. proteins and saccharides, were assigned. Fourier-transform-infrared (FTIR) spectroscopic..."

Line 181: The sentence "The differential spectrum ...." seems to be in error, as C class is more frequent by far.

**Response:** Correct, Class A and B represent less than 0.01% of active nucleation sites, not B and C.

Action Taken: We rephrased the statement.

"... shows that Class C INMs represent more than 99.9% of ..."

Line 326: This sentence is an early and incomplete statement of the results of the filtering tests.

**Action Taken:** We have removed this statement and instead elaborate on the result in detail before starting the interpretation.

Line 326-327: "We also note ..." - this is unclear.

Action Taken: We removed it.

"Individual droplets in the < 50 kDa fractions froze above -15°C due to this experiment's uncertainty."

Lines 334-335: Concentration of the suspensions combines with drop sizes to determine freezing temperature ranges.

**Response:** The statement refers to a lower sensitivity with a pL-assay. This could explain why past studies conducting filtration experiments did not see ice nucleation activity after a 100 kDa filtration since we see a decreased concentration of Class B and C INMs slightly.

Action Taken: We rephrased this statement to clarify the argumentation.

"The discrepancy can partly be explained, since both studies used pL-assays which have a lower sensitivity compared to the  $\mu$ L-assay we used."

Lines 335-336: Why does size uncertainty bring into question the 'nucleation mechanism"?

**Response:** The size uncertainty does not raise a question about the nucleation mechanism.

Action Taken: We removed it.

Lines 342-342: An example of jumping into partial conclusions.

**Response:** The statement has been removed while rephrasing the section for a more precise line of argumentation.

Line 351: " .. radii range from 240°C ... ????

**Response:** We have corrected it. It should read 240 nm.

Action Taken: "The radii range from 240 nm(non-heat-treated) to..."

Figure 5: Is Nm expressed with respect to the mass of pollen in the unfiltered sample or with respect to the mass of material left after filtration. Clearly, the latter would be more informative.

**Response:** The concentration is expressed with respect to the mass of pollen. We agree the latter would be more informative but is experimentally not easy to obtain, but we will consider this in future experiments.

Action Taken: We added the information more explicitly.

"(a) Cumulative number  $N_m(T)$  of INMs per gram of *Betula pendula* pollen of the unfiltered sample"

Line 394: Citation missing. More importantly, information on what materials are present in the samples in addition to the INMs would be welcome.

**Response:** This statement was supposed to be removed from the manuscript. Although we know that the proteinaceous *Bet v* allergens from the *Betula* species can be found on pollen grains (Schäppi et al., 1997), to the best of our knowledge, there are no publications investigating ice nucleation of the Bet v allergens.

**Action Taken:**

"... However, we know of other proteins in our sample such as the Bet v allergens, which show varying amounts of beta-sheet structures (Gajhede et al., 1996; Kofler et al., 2012; Neudecker et al., 2004; Soh et al., 2017)."

**Referee comment 2**

Comments to "Aggregation of ice-nucleating macromolecules from Betula pendula pollen determines ice nucleation efficiency" by Florian Wieland et al. (Manuscript ID: egusphere-2024-752).

Based on the twin-plate ice nucleation assay (TINA) and Vienna optical droplet crystallization analyzer (VODCA), the authors explored the ice nucleation activity of ice nucleating macromolecules (INMs) in Betula pendula pollens. The authors found three different types of INMs that caused freezing at -8.7°C, -15.7°C, and -17.4°C, respectively. The authors also found freeze drying, freeze-thaw cycles, and heat treat can alter the ice nucleation activity, which was attributed to size-varying aggregates. Overall, the science is interesting, and manuscript is written well. If the following comments can be fully addressed, the manuscript may be publishable in Atmos Chem Phys.

**Response:** We thank the referee for carefully reading our manuscript and addressing interesting points. We answer each comment in detail below.

Action Taken: Our comments are in blue, while the referee comments are in black. Parts of the manuscript are in orange, with changes and additions underlined.

**[Referee 2]**

Major comments:

**1.** The droplets in TINA experiment are shaped by the well. However, the droplets in VODCA are shaped by surrounding aqueous phase. How the different morphology and contact angles of generated droplets will influence the measured ice nucleating activity?

**Response:** The referee addresses an interesting and important aspect of ice nucleation assays. The two used assays vary in droplet size, shape, and interfaces. While this can impact freezing temperatures and sample behavior, we want to reference the study by DeMott *et al.* (2018) that compared IN measurements of soil samples with various IN assays, which showed a good match of the different assays. However, the TINA assay did not participate in this study, but the CSU-IS (Colorado State University Ice Spectrometer), which measures droplets in a PCR tray, which showed a good overlap with the results obtained using VODCA. At this point, it should be noted that TINA uses µPCR trays with smaller wells. Hence, there could still be an influencing effect that we cannot consider in this work. Nonetheless, TINA was compared to other IN assays, and Kunert *et al.* (2018) found freezing onset temperatures and similar concentrations.

This effect might not even be relevant to heterogeneous freezing of the investigated INMs from *Betula pendula* because a recent study by Bieber *et al.* (2024) suggests that *Betula pendula* INMs trigger freezing within the bulk rather than the interface. This would mean that factors such as shape and interface should not interfere with the nucleation caused by these INMs.

**Action Taken: We added the following statement to the method section:**

"Regarding differences in droplet size and shape, Miller *et al.* (2021) discuss these as potential reasons for the high background freezing temperature in detail. Looking at heterogeneous freezing we refer to Kunert *et al.* (2018), where extensive comparisons to other ice nucleation assays are reported showing similar freezing temperatures and ice nuclei concentrations. A recent study by Bieber *et al.* (2024) suggests *B. pendula* INMs trigger freezing within the bulk of the droplet rather than the surface, which means these potential effects are most likely not relevant for the investigated sample."

**2.** The authors treated the BPWW at 40 °C, 78 °C and 98 °C for 1 h or 24 h respectively. What is the relevance of such high temperatures to ambient air especially the troposphere air?

**Response:** We admit that these conditions are not relevant for ambient aerosols in the troposphere. We still think this is important data since this study tries to deepen the understanding of INMs from Betula pendula, investigating behavior under various conditions and helping us to draw conclusions on their nucleation mechanism. (see also Referee 1 – Comment 5)

Action Taken: We have made the following statements to emphasize why we did these experiments.

"Although heat treatments at high temperatures are arguably not relevant for ambient atmospheric processes, they provide valuable insight: Firstly, these treatments reveal a diverging response of the INM classes to heat, which raises further questions about possibly varying chemistry. Secondly, they are often used on field samples to determine whether the ice nuclei are of biological origin, as many biological molecules, particularly proteins, are heat-sensitive and can denature at high temperatures, losing their activity. Our data suggests a simple heat treatment is insufficient to classify whether ice nuclei are of biological origin or not."

**3.** Why the error bars of hydrodynamic radius Rh of untreated BPWW are significantly higher than other samples (Figure 6)?

**Response:** This is indeed notable but can be explained by the measurement principle of DLS, whereas a more polydisperse sample leads to larger scattering of the signals. In DLS, the signal is proportional to the particle diameter d6, meaning a small quantity of large particles can strongly influence the signals. In the untreated sample, there is a broader size distribution, which is changed during heat treatment to smaller sizes, which lowers the error bar.

Action Taken: We have added the following sentences to the manuscript.

"The untreated sample has a similar radius to the sample treated at 40°C but a much larger uncertainty (calculated from 4 independent samples). The larger uncertainty for this sample can be explained by the measurement principle of DLS, whereas the scattering signal is proportional to the sixth power of particle diameter d6. Hence, the measurements uncertainty increases drastically in polydisperse systems containing even larger particles."

**4.** Figure 1: in the left panel, the data points for pL droplets scattered significantly. One temperature corresponds to several Nm values, how to explain this?

**Response:** Our pL-assay device background starts at ~ -34°C. This means droplets containing no INMs will freeze at this temperature. The last few droplets of each of the shown dilutions are in the range of the device background. Hence, these droplets do not contain INMs.

These data points should be disregarded since the calculation of the concentration  $N_m$  does not account for the background and will result in a concentration for each data point. Data points in the device background will be around the same temperature. In this region, the dilution factor shifts the data points along the y-axis.

To avoid this confusion for future readers, we have cut off these data points from the graph. For clarification, we have already removed device-background data points from the  $\mu$ L assay. There, you would see the same effect at ~-25°C (multiple data points for a single temperature, shifted upwards on the y-axis with each dilution).